# A human pathogenic hantavirus circulates and is shed in taxonomically diverse rodent reservoirs

Samuel M. Goodfellow[1], Robert A. Nofchissey[1], Chunyan Ye[1], Jaecy K. Banther-McConnell[2], Thanchira Suriyamongkol[3], Joseph A. Cook[4], Jonathan L. Dunnum[4], Ivana Mali[5], Steven B. Bradfute[1] *

1 Center for Global Health, Department of Internal Medicine, University of New Mexico Health Sciences Center, Albuquerque, New Mexico, United States of America, 2 Department of Biology, Eastern New Mexico University, Portales, New Mexico, United States of America, 3 College of Agricultural Sciences, Southern Illinois University-Carbondale, Carbondale, Illinois, United States of America, 4 Museum of Southwestern Biology, Biology Department, University of New Mexico, Albuquerque, New Mexico, United States of America, 5 Fisheries, Wildlife, and Conservation Biology, North Carolina State University, Raleigh, North Carolina, United States of America

* sbradfute@salud.unm.edu

**Data Availability Statement:** All data is contained within the manuscript. The raw data that support the findings of this study are publicly available from

## Abstract

### Background

Orthohantaviruses are negative-sense RNA viruses that can cause hantavirus cardiopulmonary syndrome (HCPS) in humans. In the United States, Sin Nombre orthohantavirus (SNV) is the primary cause of HCPS, with a fatality rate of 36% and most cases occuring in the southwestern states. The western deer mouse, *Peromyscus sonoriensis*, is the primary reservoir for SNV; however, it remains unclear if alternative reservoirs exist.

### Results

We conducted an extensive survey of SNV genetic prevalence in wild-caught small mammal communities throughout New Mexico and observed that 27% of all animals were positive for SNV. Through longitudinal trapping at a site of patient exposure, we found that SNV circulates at a high rate in multiple species over time. Furthermore, we isolated live SNV from tissues and feces from multiple small mammal species, demonstrating infectious virus in alternative and novel reservoirs.

### Significance

Altogether, this work shows that SNV is widely prevalent and persistent throughout New Mexico in multiple small mammal reservoirs that can harbor and shed infectious virus. This encourages future work for additional surviellance efforts and revaluates host-species dynamics for New World hantaviruses.

Dryad with the identifier https://doi.org/10.5061/dryad.2547d7x2c.

**Funding:** S.M.G. was supported by UNM HSC Infectious Disease and Inflammation Program National Institutes of Health (NIH) grant T32AI007538. Eastern New Mexico University was supported through an Institutional Development Award (IDeA) - Grant Number P20GM103451; National Institute of General Medical Sciences of the National Institutes of Health. The funders had no role in study design, data collection and analysis, decision to publish, or preparation of the manuscript.

**Competing interests:** The authors have declared that no competing interests exist.

## Author summary

The first reported hantavirus cases in the United States occurred in the Four Corners region (Arizona, New Mexico, Utah and Colorado) and was found to be transmitted by the droppings of western deer mice. This led to discovery of Sin Nombre virus (SNV). Unlike hantaviruses found through Europe and Asia, SNV causes hantavirus cardiopulmonary syndrome (HCPS), which has a high mortality rate. It is generally thought that only one small mammal host is capable of carrying and transmitting hantavirus to humans. However, our group and others have found hantavirus genetic material in multiple small mammal species, but these studies have been fairly limited and did not determine if the virus could replicate in these species and be transmitted through them. Here, we conducted a large study to look for SNV in wild-caught rodents throughout New Mexico (NM) in counties with and without reported human HCPS cases. We found that SNV is widespread in all areas of NM and stably circulates in rodents at a site of human infection. Importantly, we show that multiple small mammal species can carry and shed infectious SNV. Altogether, our study provides a novel shift in the understanding of host reservoir dynamics for SNV.

## Introduction

Hantaviruses (order *Bunyavirales*, family *Hantaviridae*, genus *Orthohantavirus*) are globally distributed zoonotic viruses that have implications as a public health threat. After the initial discovery of Hantaan virus (HTNV) in 1976, additional hantaviruses have been found throughout Europe and Asia; these viruses cause hemorrhagic fever with renal syndrome (HFRS), which has a case-fatality rate of 0.1–15% and an estimated 150,000 cases annually [1–4]. The first pathogenic hantavirus in the Americas, Sin Nombre virus (SNV), was discovered in 1993 and found to cause hantavirus cardiopulmonary syndrome (HCPS), with a case-fatality rate of 36% [1,5]. Transmission occurs when humans are exposed to aerosolized viral particles from disturbance of rodent excreta [6–8]. Currently, there are no available therapeutics or vaccines for HCPS infection [9,10].

In the United States (US), SNV is the primary causative agent for HCPS, with most cases occurring throughout the west and southwest. New Mexico (NM) has 134 reported cases as of September 2024, which is more than any other state in the country. The primary reservoir is the western deer mouse (*Peromyscus sonoriensis*, previously named *P. maniculatus*) [11–13] which is widely distributed throughout North America. Within New Mexico, most human cases occur in the northwestern counties (McKinley, San Juan, and Taos), with few or no cases reported in the eastern counties of the state [14]. Interestingly, seroprevalence studies have shown orthohantaviruses prevalence in various small mammals throughout the eastern region [15]. Historically, a given orthohantavirus was thought to be spread by a single host reservoir; however, recent evidence suggests a broader host range for these viruses [16]. Our group and others have recently found that additional small mammals other than *P. sonoriensis* harbor SNV genetic material [17–19]. In the original identification of *P. sonoriensis* as the primary SNV reservoir, several rodent species were also found to be positive by serology or PCR but these records were characterized as spillover events [13]. No studies have analyzed whether small mammals other than *P. sonoriensis* can act as SNV reservoirs by carrying and shedding live virus.

Past surveillance efforts investigating SNV prevalence in New Mexico have not been distributed evenly throughout the state, with the largest efforts only serology-based [15,20,21].

Although there are no known human SNV cases in eastern New Mexico, a recent study by our group demonstrated that the virus is present in small mammals in that region [17]. Additional surveillance efforts throughout the entire state in counties with and without reported HCPS cases are needed to compare prevalence of circulating SNV within small mammal populations.

In this study, we screened 1,561 wild-caught small mammals across New Mexico to test whether SNV is circulating among multiple syntopic species in these rodent communities. We also performed a longitudinal prevalence study at a site of patient exposure. We report that non-*P. sonoriensis* small mammals are reservoirs for SNV, with consistent circulation of virus in rodent populations over time, providing key information regarding the SNV host-virus dynamics and distribution.

## Results

### SNV prevalence in wild-caught small mammals throughout New Mexico

A total of 1,471 small mammals were captured between 2019 to 2023 from late winter to late fall throughout New Mexico. Specimens captured from Eastern New Mexico University (ENMU) used only lung tissue for SNV testing. For specimens captured from the University of New Mexico (UNM), lungs and other tissues were collected, and lungs were tested for SNV. An additional 90 *P. sonoriensis* lung tissues from animals captured in 1994 and 1998 were tested from lung tissue archives at the Museum of Southwestern Biology (MSB). From these 1561 samples, Cricetidae (New World rats and mice, hamsters, voles and/or relatives) accounted for 1,086 (69%) of all captured small mammals, with Heteromyidae (i.e., kangaroo rats, pocket mice) comprising 284 (18%), Sciuridae (squirrels, chipmunks, prairie dogs) with 121 (8%), Muridae (black rats, house mice) with 51 (3%), Geomyidae (gophers) with 18 (<1%), as well as 1 member (<1%) of the family Procyonidae (coatis, racoons, and/or relatives). The most abundant rodent captured was *P. sonoriensis* (n = 452), followed by *Onychomys leucogaster* (Northern Grasshopper Mouse, n = 166) and *Dipodomys merriami* (Merriam's Kangaroo Rat, n = 116). Trapping was performed across various terrains ranging from cold to warm deserts and central arid prairies.

We found that 415 (26%) of the 1,561 tissue samples were positive for SNV viral RNA by qPCR (Table 1 and Fig 1A). To determine if SNV could be detected in tissue archives, we screened 90 frozen lungs and found 23 positives through RT-qPCR (Fig 1B), highlighting the potential of biorepositories for pathogen detection [22]. *Peromyscus sonoriensis* accounted for 165 (40%) of all positives, followed by *O. leucogaster* with 40 (10%), *Peromyscus truei* (Pinyon Mouse) with 28 (7%), and *P. leucopus* (White-footed Mouse) with 24 (6%). Several small mammal species have been shown to have anti-hantavirus antibodies or SNV genetic material in previous surveillance studies in the US [13,17,18,20,21,23]. We found SNV in potential novel hosts, including *Peromyscus laceianus* (Northern White-Ankled mouse), which is restricted to northern Mexico and the southwestern US (New Mexico, Texas and Oklahoma), *Reithrodontomys montanus* (Plains harvest mouse), *Neotoma leucodon* (white-toothed woodrat), *Dipodomys spectabilis* (banner-tailed kangaroo rat), and *Thomomys bottae* (Botta's pocket gopher) and *talpoides* (Northern pocket gopher) (Table 1).

Because multiple *Peromyscus* species are capable of experimental SNV infection and replication [19], we compared SNV RNA copies in wild *P. sonoriensis* against other wild caught *Peromyscus* species). We found no statistically significant differences in SNV RNA copy numbers among different *Peromyscus* species (Fig 2A). We then compared *P. sonoriensis* to all other positive small mammals to determine if SNV RNA viral copies differed. We observed a significant difference between *P. sonoriensis* and non-*P. sonoriensis*, with a higher copy number in *P. sonoriensis* (Fig 2B). We then compared *P. sonoriensis* directly with other non-

**Table 1. Overview of all small mammals screened for SNV viral RNA by RT-qPCR including archived museum tissue samples.** Tissue represents lung tissue from a majority of rodents; however approximately 3.3% of all the specimens (51 total in Rio Arriba and Lincoln counties) had only liver or kidney available for testing. Shown is the number of each genera screened and SNV+ prevalence.

| Species | Common name | # Screened (% total) | Tissue SNV + % (+ves/total tested) |
|---|---|---|---|
| **Cricetidae** | | | |
| *Peromyscus sonoriensis* | Western Deer Mouse | 452 (29%) | 36% (165/452) |
| *Peromyscus leucopus* | White-footed Mouse | 92 (6%) | 25% (24/92) |
| *Peromyscus truei* | Pinyon Mouse | 103 (7%) | 27% (28/103) |
| *Peromyscus boylii* | Brush Mouse | 28 (2%) | 28% (9/28) |
| *Peromyscus eremicus* | Cactus Mouse | 16 (1%) | 25% (5/16) |
| *Peromyscus nasutus* | Northern Rock Mouse | 26 (2%) | 15% (4/26) |
| *Peromyscus laceianus* | Northern White-Ankled Mouse | 7 (0.5%) | 28% (2/7) |
| *Neotoma mexicana* | Mexican Woodrat | 20 (1%) | 4% (1/20) |
| *Neotoma albigula* | White-throated Woodrat | 22 (1%) | 4% (1/22) |
| *Neotoma stephensi* | Stephen's Woodrat | 1 (0.2%) | 0% (0/1) |
| *Neotoma cinerea* | Bushy-tailed Woodrat | 2 (0.2%) | 50% (1/2) |
| *Neotoma micropus* | Southern Plains Woodrat | 64 (4%) | 16% (10/64) |
| *Neotoma leucodon* | White-toothed Woodrat | 43 (3%) | 19% (9/43) |
| *Reithrodontomys megalotis* | Western Harvest Mouse | 14 (0.9%) | 36% (5/14) |
| *Reithrodontomys montanus* | Plains Harvest Mouse | 11 (0.7%) | 18% (2/11) |
| *Microtus longicaudus* | Long-tailed Vole | 4 (0.3%) | 25% (1/4) |
| *Microtus montanus* | Montane Vole | 6 (0.5%) | 17% (1/6) |
| *Onychomys leucogaster* | Northern Grasshopper Mouse | 166 (11%) | 23% (40/166) |
| *Sigmodon hispidus* | Hispid Cotton Rat | 8 (0.5%) | 12% (1/8) |
| *Baiomys taylori* | Northern Pygmy Mouse | 1 (0.2%) | 0% (0/1) |
| **Sciuridae** | | | |
| *Neotamias minimus* | Least Chipmunk | 31 (2%) | 32% (10/31) |
| *Neotamias quadrivittatus* | Colorado Chipmunk | 75 (5%) | 13% (10/75) |
| *Neotamias dorsalis* | Cliff Chipmunk | 7 (0.4%) | 33% (2/7) |
| *Tamiasciruis hudsonicus* | American Red Squirrel | 3 (0.2%) | 0% (0/3) |
| *Cynomys gunnisoni* | Gunnison's Prairie Dog | 1 (0.2%) | 0% (0/1) |
| *Sorex navigator* | Western Water Shrew | 1 (0.2%) | 0% (0/1) |
| *Callospermophilus lateralis* | Gold-mantled Ground Squirrel | 1 (0.2%) | 100% (1/1) |
| *Otospermophilis variegatus* | Rock Squirrel | 2 (0.2%) | 0% (0/2) |
| **Heteromyidae** | | | |
| *Dipodomys merriami* | Merriam's Kangaroo Rat | 116 (8%) | 16% (18/116) |
| *Dipodomys ordii* | Ord's Kangaroo Rat | 64 (4%) | 22% (17/64) |
| *Dipodomys spectabilis* | Banner-tailed Kangaroo Rat | 3 (0.2%) | 67% (2/3) |
| *Chaetodypus intermedius* | Rock Pocket Mouse | 3 (0.2%) | 0% (0/3) |
| *Chaetodipus eremicus* | Chihuahuan Pocket Mouse | 13 (0.4%) | 8% (1/13) |
| *Chaetodipus hispidus* | Hispid Pocket Mouse | 16 (0.4%) | 38% (6/16) |
| *Perognathus flavus* | Silky Pocket Mouse | 65 (4%) | 26% (17/65) |
| *Perognathnus merriami* | Merriam's Pocket Mouse | 3 (0.2%) | 33% (1/3) |
| *Perognathus flavescens* | Plains Pocket Mouse | 1 (0.2%) | 0% (0/1) |
| **Muridae** | | | |
| *Mus musculus* | House Mouse | 50 (3%) | 30% (15/50) |
| *Rattus norvegicus* | Brown Rat | 1 (0.2%) | 0% (0/1) |
| **Geomyidae** | | | |

*(Continued)*

**Table 1.** (Continued)

| Species | Common name | # Screened (% total) | Tissue |
| | | | SNV + % (+ves/total tested) |
| --- | --- | --- | --- |
| *Thomomys bottae* | Botta's Pocket Gopher | 13 (1%) | 38% (5/13) |
| *Thomomys talpoides* | Northern Pocket Gopher | 4 (0.2%) | 25% (1/4) |
| *Geomys arenarius* | Desert Pocket Gopher | 1 (0.2%) | 0% (0/1) |
| **Procyonidae** | | | |
| *Procyon lotor* | Racoon | 1 (0.2%) | 0% (0/1) |
| Total(s) | | 1561 (100%) | 26.6% (415/1561) |

*Peromyscus* species. We found that *P. sonoriensis* had significantly higher copy numbers compared to *Mus musculus* (common house mouse), but not to *T. bottae* or *Neotamias* species (Fig 2C).

Next, we tested whether the prevalence of SNV in wild-caught small mammals varied by geographic location to examine the correlation with the incidence of reported HCPS cases in New Mexico. As expected, we found SNV in small mammals in counties with previously reported HCPS cases; however, we also found SNV in small mammals in counties with no reported HCPS cases, demonstrating that the virus circulates in areas with no reported human

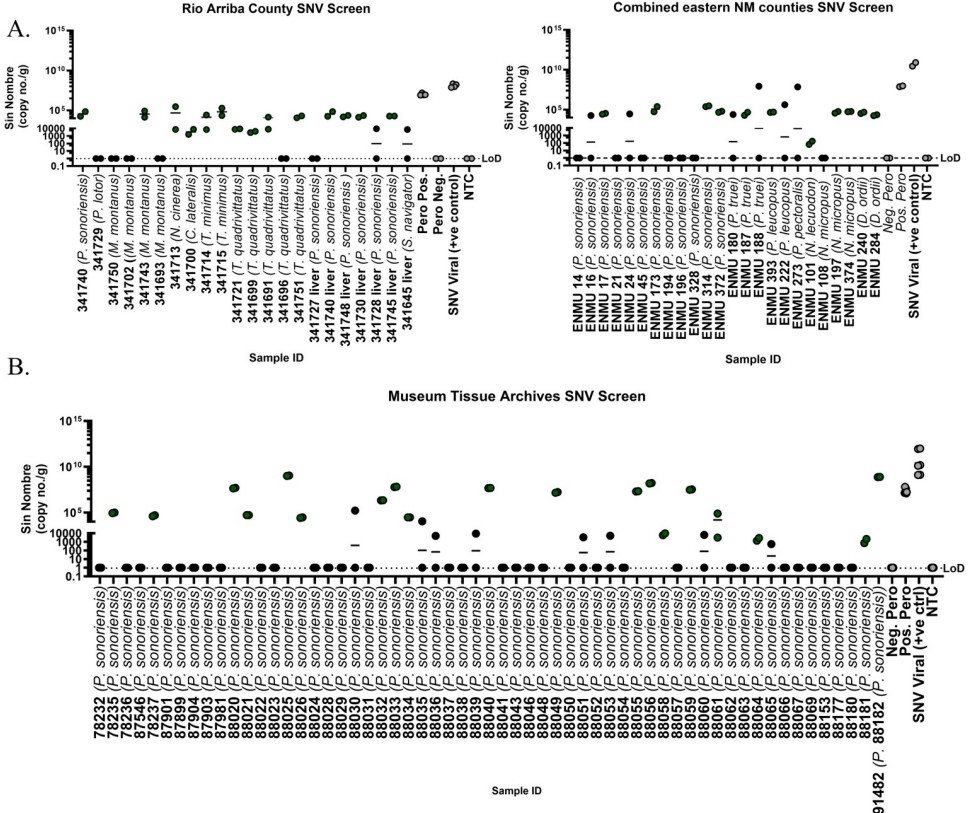

**Fig 1. Detection of SNV in wild-caught small mammals and archival museum samples.** Singleplex SNV-specific amplification of SNV from lungs of wild-caught small mammals from various counties in NM (A) and frozen collected tissue (B). Limit of detection (LoD) was determined using a no-template control (NTC) and SNV-negative rodent lung tissue (Neg. Pero). Panels A and B show selected screens. Samples collected through the MSB are labeled with MSB: Mamm ID.

A

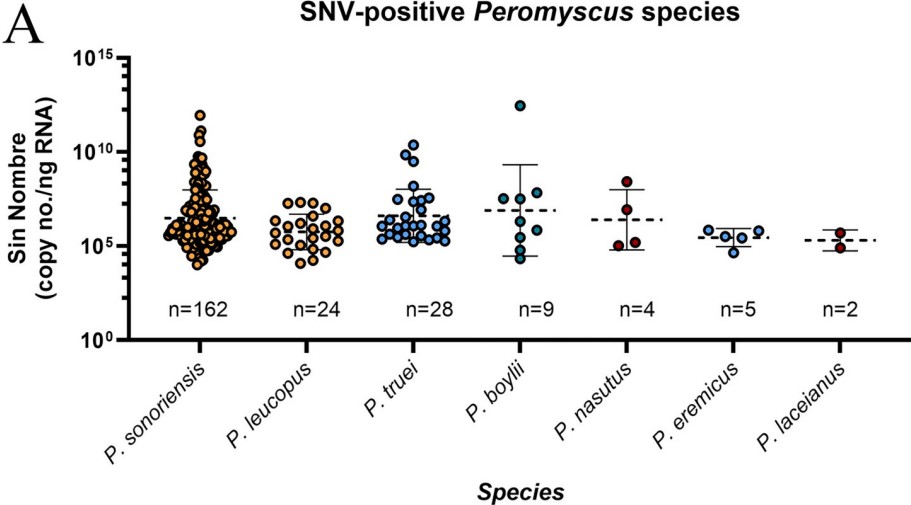

B

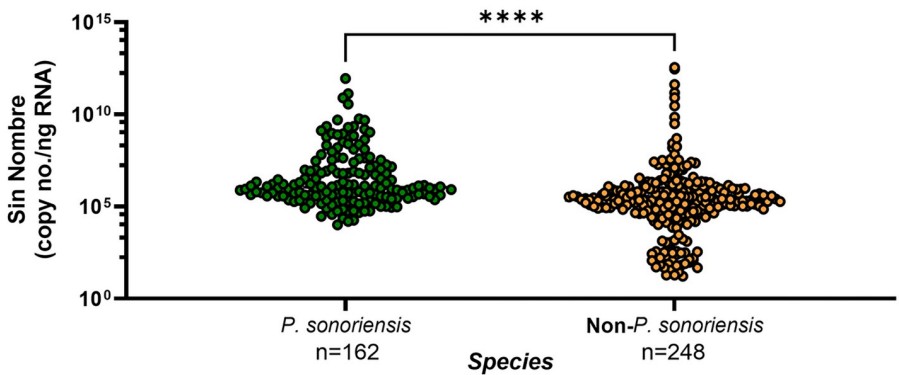

C

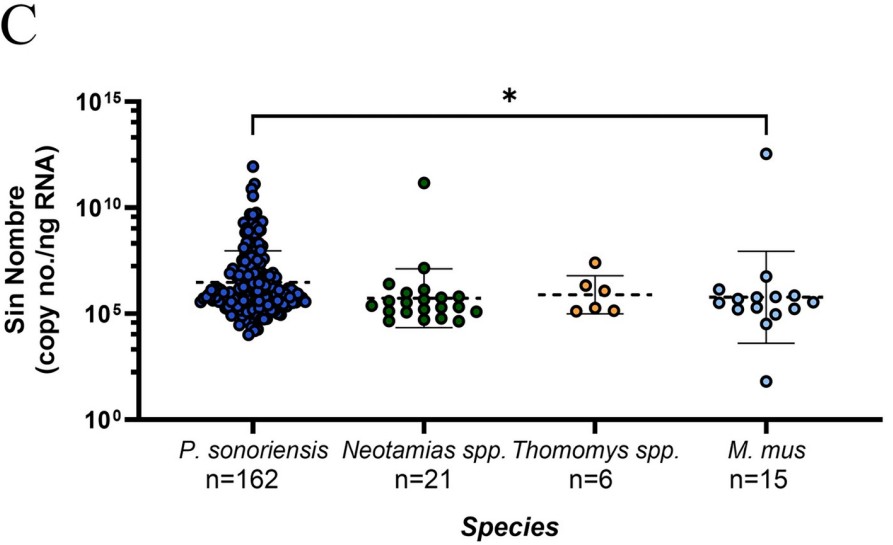

**Fig 2. Detection of SNV in several small mammal species.** Copies per gram of SNV viral RNA detected was calculated and plotted. (A) Positive-*Peromyscus* species were compared. (B) Positive *P. sonoriensis* versus non-*P. sonoriensis* small mammals are shown. (C) *P. sonoriensis* along with example rodent species positive for SNV. Geometric mean is shown in graphs. Mann-Whitney and Kruskal-Wallis tests were used to compare values. * indicates a P value of <0.01 and **** indicates a P value of <0.0001. n indicates the total number of specimens in that column.

infections (Fig 3A and data previously reported by our group [7]). Although significant viral RNA copy numbers were found in both sets of counties (Fig 3B), we found that small mammals in counties with HCPS-reported cases had higher lung SNV copy numbers than those trapped in counties without known human cases (Fig 3C). We also compared *P. sonoriensis* SNV-positive samples across counties but found no significant differences in viral RNA copy numbers in these animals (Fig 3D).

## Longitudinal study at site associated with patient exposure

In 2020, we trapped small mammals at two sites of potential exposure for an HCPS patient. One site was outside the patient's residence (Site 1) and the second site was ~22 km distant

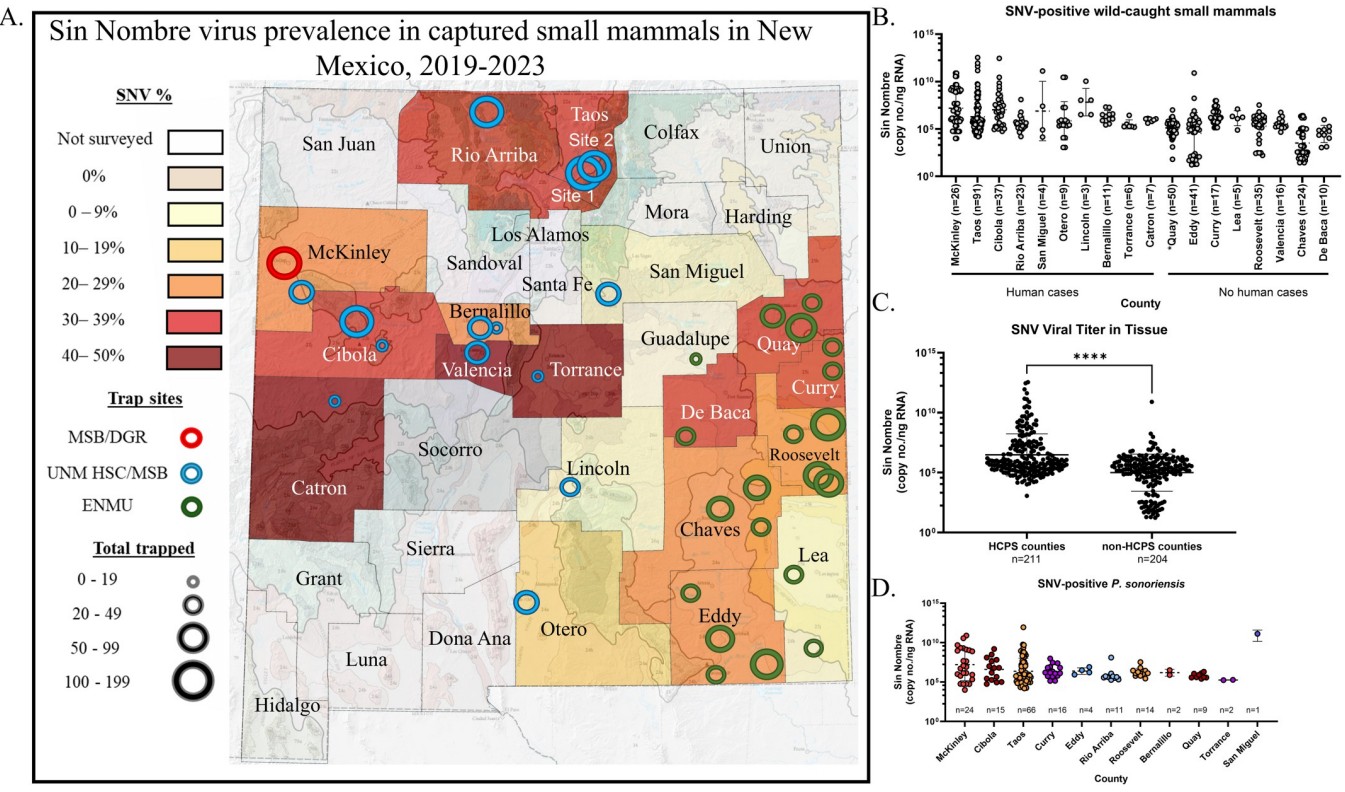

**Fig 3. Detection of SNV viral RNA in small mammals throughout New Mexico irrespective of human reported cases.** (A) Heat map of SNV-positive small mammals across New Mexico by county. Survey sites are shown as circles by location, collectors, and by number of total rodents trapped per site. Museum of Southwestern Biology (red) University of New Mexico Health Sciences Center/Museum of Southwestern Biology (blue) Eastern New Mexico University (green). Samples from Eastern New Mexico University have been previously reported (Banther-McConnell et al. 2024 [24]). Copy number per ng of SNV viral RNA detected was calculated and plotted. (B) SNV-positive small mammals across 19 counties is shown. * indicates that during peer review of this publication, the first hantavirus human case was reported in Quay County August 2024. (C) Total-positive small mammals compared between counties with HCPS-reported cases and without is shown. (D) Positive-SNV *P. sonoriensis* copies by county is shown. *n* listed shows the total small mammals per category. Geometric mean and standard deviation is shown. Mann-Whitney t-test was used with **** indicates a P value of <0.0001. Level III and IV Ecoregion GIS data was sourced from the freely available United States Environmental Protection Agency–Ecoregion database (https://www.epa.gov/eco-research/level-iii-and-iv-ecoregions-state) (see EPA Accessibility Statement at https://www.epa.gov/accessibility/epa-accessibility-statement) and heat map of counties was created with a freely available software, MapChart.net.

**Table 2. Summary of longitudinal small mammal capturing and testing for SNV viral RNA in lung tissue at two sites in Taos county.** Trapping and screening was conducted each August from 2020–2023. Site 1 and Site 2 are labeled in this table and shown in Fig 3; sites and 2020 data are previously described in [18].

| Species | # Captured (%) | Location #—Site Trapped | Lung tissue (% SNV +) | | | | |
| --- | --- | --- | --- | --- | --- | --- | --- |
| | | | Total SNV + # (%) | 2020 # (%) | 2021 # (%) | 2022 # (%) | 2023 # (%) |
| *Peromyscus sonoriensis* | 151 (59) | 118 –Site 1 | 55 (47) | 32 (41) | 21 (43) | 40 (38) | 25 (72) |
| | | 33 –Site 2 | 11 (33) | 9 (44) | 8 (13) | 11 (36) | 5 (40) |
| *Peromyscus boylii* | 8 (3) | 0 –Site 1 | - | - | - | - | - |
| | | 8 –Site 2 | 2 (25) | 4 (50) | 1 (0) | 2 (0) | 1 (0) |
| *Peromyscus truei* | 23 (9) | 0 –Site 1 | - | - | - | - | - |
| | | 23 –Site 2 | 5 (22) | - | 3 (33) | 10 (10) | 10 (30) |
| *Mus musculus* | 23 (9) | 23 –Site 1 | 8 (35) | 9 (33) | - | 6 (50) | 8 (25) |
| | | 0 –Site 2 | - | - | - | - | - |
| *Neotoma mexicana* | 16 (6) | 0 –Site 1 | - | - | - | - | - |
| | | 16 –Site 2 | - | 4 (0) | 6 (0) | 5 (0) | 1 (0) |
| *Neotoma albigula* | 2 (<1) | 0 –Site 1 | - | - | - | - | - |
| | | 2 –Site 2 | - | - | - | 1 (0) | 1 (0) |
| *Neotoma cinerea* | 1 (<1) | 0 –Site 1 | - | - | - | - | - |
| | | 1 –Site 2 | - | - | - | 1 (0) | - |
| *Neotamias minimus* | 22 (8) | 5 –Site 1 | 2 (40) | 5 (40) | - | - | - |
| | | 17 –Site 2 | 5 (29) | 6 (33) | 3 (33) | 6 (33) | 2 (0) |
| *Neotamias quadravittius* | 4 (2) | 0 –Site 1 | - | - | - | - | - |
| | | 4 –Site 2 | 1 (25) | - | 1 (100) | 3 (0) | - |
| *Reithrodontomys megalotis* | 3 (1) | 1 –Site 1 | - | - | - | - | - |
| | | 2 –Site 2 | 1 (50) | - | - | - | 3 (33) |
| *Sigmodon hispidus* | 1 (<1) | 1 –Site 1 | 1 (100) | - | - | 1 (100) | - |
| | | 0 –Site 2 | - | - | - | - | - |
| *Cynomys gunnisoni* | 1 (<1) | 1 –Site 1 | - | - | - | 1 (0) | - |
| | | 0 –Site 2 | - | - | - | - | - |
| Site 1 Captures (%) | | | 66 (73) | 46 (39) | 22 (36) | 48 (40) | 34 (62) |
| Site 2 Captures (%) | | | 25 (27) | 23 (35) | 21 (19) | 39 (18) | 22 (27) |
| Total(s) | 255 (100) | | 91 (36) | 69 (38) | 43 (28) | 87 (30) | 56 (48) |

(Site 2). Using SNV genome sequencing of patient samples and trapped rodents, we previously identified Site 1 as the likely site of infection [18]. To examine whether SNV circulation rates in the small mammal community persisted over time, we continued to sample these two sites in August of 2020, 2021, 2022, and 2023. Most small mammals caught at these sites were *P. sonoriensis* (59%), followed by *Neotamias minimus* (Least Chipmunk, 10%), *P. truei* (9%) and *M. musculus* (9%). We also found *Sigmodon hispidus* (Hispid Cotton Rat) at Site 1, a potentially new distributional extension for this species in Taos county of New Mexico. Overall, we found that 91 of the 255 small mammals (36%) were positive for SNV genomic RNA. In 2020, 38% of rodents were SNV positive, which dipped to 28% in 2021 and 30% in 2022, followed by a rebound to 48% in 2023. We observed maintenance of SNV circulation across multiple rodent species at both sites across all four years at the likely site of exposure (Site 1) (Table 2).

## Viral isolation and validation from alternative host reservoirs for SNV

After discovering that multiple small mammal species were positive for SNV viral RNA, we attempted to isolate live viruses from these potential hosts to verify if they can serve as reservoirs. Previous attempts to isolate hantaviruses using Vero E6 cells have been challenging [24]. Additionally, few studies have attempted live SNV isolation from feces or salivary glands, which are suspected modes of transmission. To that end, we established a protocol to isolate

live SNV from tissue homogenates of infected *P. sonoriensis* by sequential weekly passaging on *P. sonoriensis* cells (pulmonary microvascular endothelial cells–PMVEC) [25]. We found that we can isolate live SNV from small mammal tissues with this method (S1 Fig). Using this approach, we analyzed whether live SNV can be isolated from tissue homogenates, feces, and salivary glands from *P. sonoriensis* and non-*P. sonoriensis* tissues.

We attempted to isolate live SNV from 90 samples, 43 tissue homogenates (either lung only, or combined lung, liver, kidney and/or heart), 23 feces, and 24 salivary glands from *P. sonoriensis*, *Neotamias* species, *Thomomys* species, *M. musculus*, *P. leucopus*, and other small mammal species. We selected samples that had low Ct values and represented different species to determine if multiple species could carry and shed live SNV. After 3–5 passages, we detected SNV from tissue homogenates in five of six (83%) *P. sonoriensis*, 4/7 (57%) *Neotamias species*, 5/6 (83%) *M. musculus*, 4/6 (67%) *P. leucopus*, and 4/4 (100%) *Thomomys* species (Fig 4 and Table 3).

Next, we tested whether live virus could be isolated from feces and salivary glands of different small mammal species because SNV reservoirs transmits virus through excreta and saliva (Fig 5 and Table 3). For excreta, we isolated live virus from three of four isolate attempts (75%) for *P. sonoriensis* and from all seven *Neotamias* species. We also detected viral RNA in at least one isolate from *T. bottae*, *M. musculus*, *P. leucopus*, *P. boylii, and S. hispidus*. For salivary glands, we isolated live SNV from *P. sonoriensis* (5/6, 83%), *Neotamias* species (3/4, 75%), *P. leucopus* (3/3, 100%), *M. musculus* (2/3, 67%), *P. truei* (3/6, 50%), *P. boylii* (1/1) and *S. hispidus* (1/1).

## Discussion

Surveillance of hantavirus host reservoirs in areas with both documented and undocumented HCPS cases is critical to understanding viral circulation dynamics, preventing and mitigating outbreaks, and enhancing diagnostic strategies [26]. Previous hantavirus surveillance in North America has relied on cost-effective serological approaches [13,15,27–31]; however, cross-reactivity of antibodies across multiple hantavirus species limits the ability for identification [13,32]. The advantages of the sensitivity and specificity of RT-qPCR are well-known [33–36] and allows for detection of early-stage infection (before development of antibodies) and detection of low viral load, which is often observed in hantavirus infections [37–39]. In this study, new insights into dynamics and extent of circulation of SNV in New Mexico small mammal populations fundamentally changes our perspective on host-reservoir distribution for SNV.

We observed that 26.6% of 1,561 small mammals were positive for SNV genomic material (Table 1); previous small mammal seroprevalence studies have reported rates between 2.8% to 51.3% [13,17,27–31]. Natural hantavirus infections in rodents are thought to be chronic, with little to no pathologic effects or signs of disease [40]. During infestations at sites of human infection, a higher chance of capturing positive infected animals can occur but is variable. For example, in Utah, 29.7% antibody prevalence in the rodent populations as a consequence of human disturbance was reported [30]. Another long-term study was performed at Yosemite where seroprevalence varied based on year or location between 4.8 to 14.5%. Lastly, an earlier study was conducted in national parks in regions that had no reported human SNV cases that showed antibody prevalence in 7% of deer mice populations, but also was dependent on location, where northeast regions had the highest prevalence [20]. Altogether, the variability between studies appears to be dependent on location, rodent populations and species diversity, along with frequency of human spillover [41]. Discovery of live SNV beyond the known reservoir *P. sonoriensis*, to multiple rodent species spanning across four phylogenetically diverse families (Cricetidae, Sciuridae, Heteromyidae, Muridae) alters the long-standing

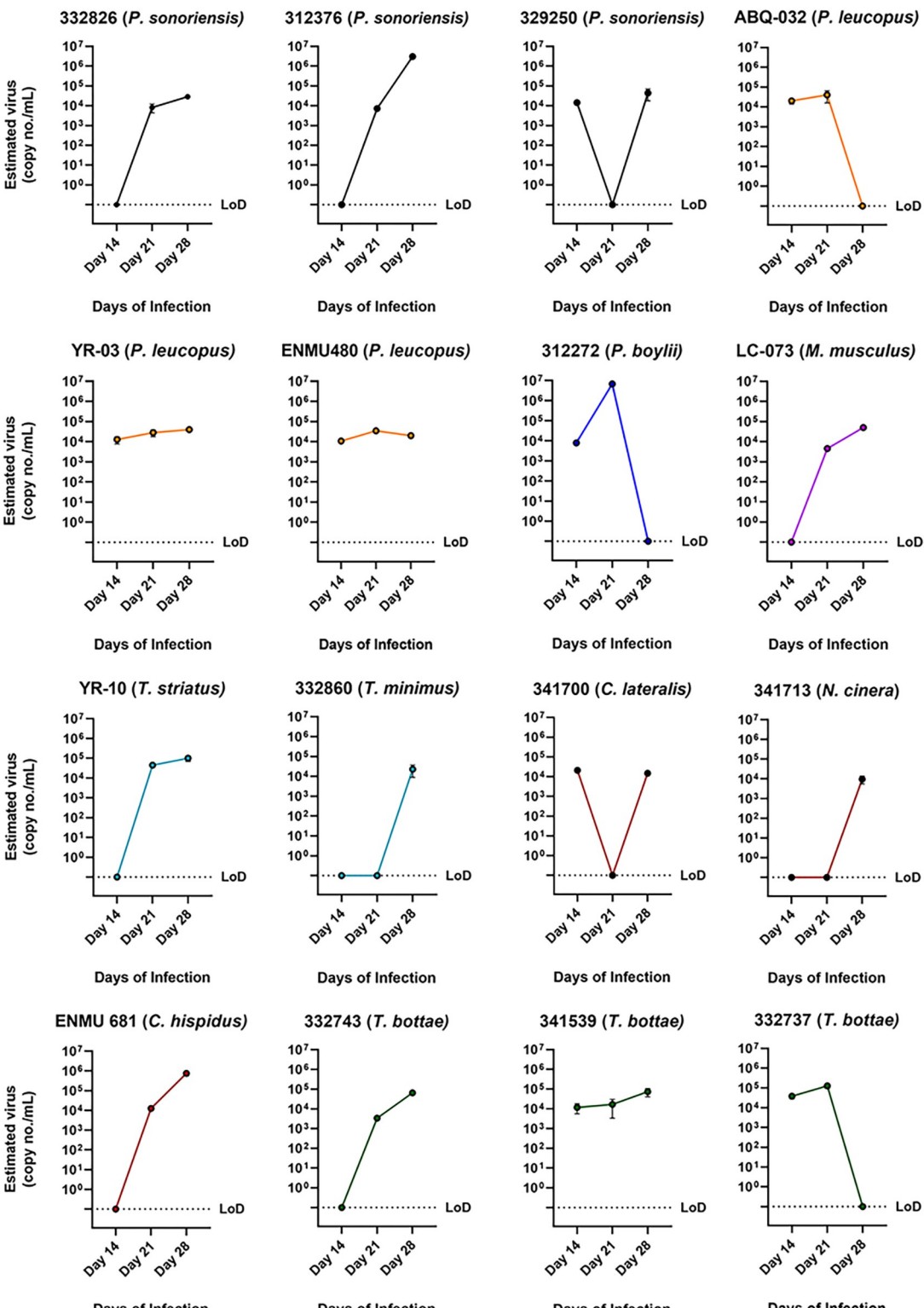

**Fig 4. Isolation of hantavirus using SNV-positive tissue homogenates in PMVECs.** Tissue homogenates of either lung, kidney, liver and/or heart were used to infect PMVEC cells. Time points were taken over several days and detected viral RNA by PCR. Limit of detection (LoD) was determined by No template control (NTC) and control supernatant from uninfected cells.

**Table 3. Summary of attempted hantavirus infections and isolates using PMVECs.** A positive sample was determined by at least one day of detection for SNV viral RNA. *Misc.* indicates additional small mammal species (*Reithrodontomys megalotis* [positive], *Callospermophilus lateralis* [positive], *Microtus montanus*, *Dipodomys spectabilis*, and *Perognathus flavus*).

| Species | +SNV RNA (tissue) | +SNV RNA (feces) | +SNV RNA (salivary glands) |
|---|---|---|---|
| *Peromyscus sonoriensis* | 5/6 (83%) | 3/4 (75%) | 5/6 (83%) |
| *Peromyscus leucopus* | 4/6 (67%) | 2/4 (50%) | 3/3 (100%) |
| *Peromyscus boylii* | 1/1 (100%) | 2/2 (100%) | 1/1 (100%) |
| *Peromyscus truei* | 1/1 (100%) | - | 3/6 (50%) |
| *Neotamias spp.* | 4/7 (57%) | 7/7 (100%) | 3/4 (75%) |
| *Chaetodipus hispidus* | 2/2 (100%) | - | - |
| *Thomomys spp.* | 4/4 (100%) | 1/1 (100%) | - |
| *Sigmodon hispidus* | 1/1 (100%) | 1/1 (100%) | 1/1 (100%) |
| *Mus musculus* | 5/6 (83%) | 3/4 (75%) | 2/3 (67%) |
| *Neotomas spp.* | 3/5 (60%) | - | - |
| *Misc.* | 2/5 (40%) | - | - |

coevolutionary paradigm that each hantavirus has a single primary host. Although multiple studies have detected hantavirus antibodies or genetic material in several small mammal species, these were considered spillover events that are not significantly involved in SNV spread or infection [13].

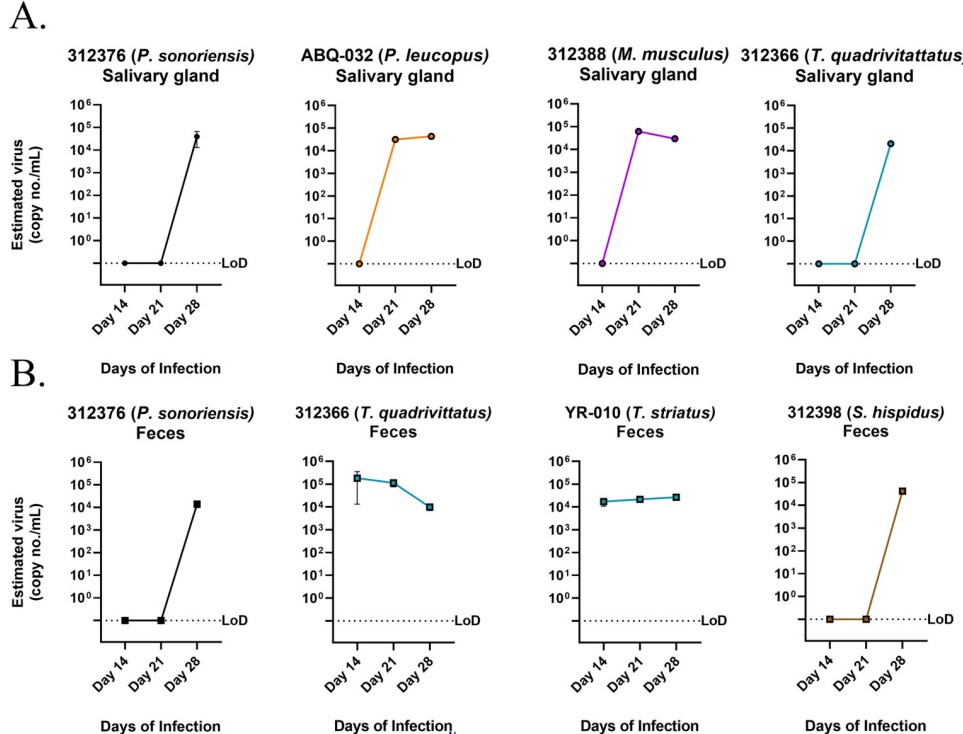

**Fig 5. Isolation of hantavirus by SNV-positive rodent salivary glands and feces.** (A) Salivary glands were homogenized and used for infection. (B) Feces from the colon were homogenized and treated to infect PMVECs. Time points were taken over several days and viral RNA was detected by PCR. The limit of detection (LoD) was determined by No template control (NTC) and control supernatant from uninfected cells.

Importantly, we show here that SNV genomic material is detected in multiple small mammal species in regions that have not reported any human HCPS cases. Although New Mexico has the highest number of reported HCPS cases in the US, the distribution within the state is concentrated in the northwest region. We found that small mammals trapped in several counties in eastern New Mexico (Eddy, Curry, Roosevelt, and Lea) where no human HCPS cases have been reported have high SNV copy numbers, suggesting the possibility that human infection could occur in these areas, although this requires further investigation (Fig 3).

Furthermore, we demonstrate that SNV circulates for years in small mammal hosts at sites associated with patient infection. The primary trapped species at this site was *P. sonoriensis*, which accounted for 59% of the samples captured, with an SNV infection range of 38% to 72% each year. This is a significant finding, as previous longitudinal persistence studies have mostly been based on seroprevalence or were not conducted directly at known sites of human infection [27,29,42,43].

One challenge to hantavirus studies has been the difficulty in isolating viruses from infected tissues, as SNV is often present at low levels in patients or small mammal reservoirs. Previous efforts at isolating SNV have used Vero E6 cells, which lack type I IFN, to isolate live virus [44], but this process is cumbersome and inefficient. We show here that *P. sonoriensis* pulmonary microvascular endothelial cells [25] can be used to isolate live SNV from tissue samples. We found that SNV genetic material and live virus was found in not only *P. sonoriensis*, but in several other small mammal species. Importantly, we isolated live SNV from tissues as well as from salivary glands and feces, suggesting that SNV can replicate and shed from non-*P. sonoriensis* species. These studies are the first to directly detect live SNV in small mammals other than *P. sonoriensis*, demonstrating that multiple SNV reservoirs exist. Further studies need to be conducted to determine whether SNV isolated from non-*P. sonoriensis* is capable of causing HCPS in humans. Furthermore, sequencing of full genomes from these samples is necessary to establish 1) if different strains are found in different regions of New Mexico; 2) if there are viral mutations found in humans that are not present in small mammal hosts; and 3) whether there are significant differences between SNV found in different small mammal hosts.

Altogether, our major findings can be summarized into three points. First, we show that SNV is prevalent in small mammals in regions with no known HCPS-reported cases. Second, SNV remains persistent in circulation in small mammals at sites of patient exposure for years. Lastly, multiple small mammal species in addition to *P. sonoriensis* are capable of harboring and shedding live SNV. This work expands our current knowledge of natural known reservoirs. Furthermore, finding SNV in small mammals in areas where no human HCPS cases have been reported may suggest that human infections are underreported or that there are differences in transmission or pathogenicity of SNV in these areas. This work demonstrates the necessity for performing additional longitudinal and targeted surveillance that can contribute to predictive models of orthohantaviral outbreaks, as done for Seoul virus [45].

## Materials and methods

### Ethics statement

All field procedures were performed following the animal care and use guidelines of the American Society of Mammalogists [46,47] and approved by the University of New Mexico Institutional Animal Care and Use Committee, collected under the New Mexico Department of Game and Fish permit (Authorization number 3300). Eastern New Mexico University samples were collected through the approval of New Mexico State University protocol #2019–016 and New Mexico Department of Game and Fish permit (Authorization number 3621).

## Plasmid construction and cultured virus

A plasmid carrying Sin Nombre virus N was generated by GenScript using the N gene from the S segment sequence in a pFastBac1 backbone and was used for the RT-qPCR standard curve. For positive virus controls, Vero E6 cells were infected with an MOI of 1 with Sin Nombre virus (SN77734) in a BSL-3 facility and RNA was extracted after lysis.

## Rodent sample collection

A total of 1,471 small mammals were collected over four years (2019–2023) between the University of New Mexico (UNM) and Eastern New Mexico University (ENMU). An additional 90 small mammal snap-frozen lung tissues were tested from the Museum of Southwestern Biology ARCTOS and DGR repository. Sherman live traps (3 x 3.5 x 9in H.B. Sherman Co., Tallahassee, Florida) were used and baited with peanut butter and oats. Holistic museum specimens were prepared according to best practices for emerging pathogen research and data-based in a relational collection management system (Arctosdb.org) to facilitate linkage of host specimen data and derived pathogen data [22,48]. Standard measurements (total length–tail length–hind foot (with claw)–ear (from notch)–weight), reproductive data (sex, reproductive status, testes, and embryo crown rump measurements), and age were recorded. Species identifications were determined through a combination of measurement data and morphological characters while some samples were confirmed through cytochrome b sequence analysis. Tissues collected were snap frozen and included brown fat, spleen, heart, lung, kidney, liver, colon (with feces), urinary bladder (with urine if present), salivary glands, muscle, and serum from blood centrifuged in the field. Specimens were also screened for ectoparasites. Animal specimens collected from UNM and ENMU were deposited in the University of New Mexico, Museum of Southwestern Biology (MSB) and Eastern New Mexico University Natural History Museum.

## RNA extraction

RNA extraction of small mammal tissue was performed using the QIAmp Viral RNA Mini Kit (Qiagen) following the manufacturer's instructions with slight modifications in suggest tissue input and buffer usage. Briefly, an average of 40 mg of frozen tissue was added to a bead beater tube preloaded with 1.0 g of 1.0 mm dia. Zirconia beads (BioSpec Cat. No. 1107911zx), 1.0 g 2.0 mm dia. Zirconia beads (BioSpec Cat. No. 11079124zx) and 800 μl of AVL buffer. The tissue was bead beat using a Benchmark Bead Bug-6 at a speed of 4350 rpm for 30 seconds for 1 cycle. Homogenates were centrifuged, pipetted into a microfuge vial, centrifuged to remove debris, then the clear lysate was pipetted into a fresh microfuge vial. The RNA carrier was added to the clear lysate and the RNA extraction proceeded as per the manufacturer's instructions.

## Quantitative reverse transcriptase polymerase chain reaction (RT-qPCR)

Two-step reverse transcription (RT) using SuperScript II (Invitrogen, Thermo Fisher Scientific) was established for the QuantStudio5 series (Applied Biosystems). The RT was performed using 5 μl of RNA (~500 ng) with 1 μl of SuperScript II containing 4 μl 5x First Strand Buffer, 2 μl 0.1M DTT, 1μl RNaseOUT, 1 μl random primers, 1 μl dNTP Mix (10 mM), and 5 μl of RT-qPCR grade water. This reaction was incubated at 65°C for 5 minutes, placed on ice briefly, followed by 10 minutes at room temperature for binding with a 50-minute reaction at 42°C, then terminated by 15 minutes at 70°C. RT-qPCR reactions were carried out by using 10 μl TaqMan Fast Advanced Master Mix (Applied Biosystems, Thermo Fisher Scientific Catalog #

4444965) containing 2 μl cDNA (~200 ng); 2 μl primer (2.5 uM) 1 μl of either probe (10uM) and 4–5 μl DEPC-treated water (Nalgene) in a 20 μl final volume reaction. For each sample, duplicate or triplicate wells were tested using the following cycling conditions: 20 seconds at 95˚C for the hold stage, while the PCR stage used a 1 second at 95˚C followed by 52˚C for 20 seconds for a total of 40 cycles. Controls included no template control (NTC), positive SNV viral sample, plasmid used for standard curve, positive and negative patient sample or wild-caught rodent samples. Additionally, β-actin primers (F–ATG TAC GTA GCC ATC CAG GC; R–TCT TGC TCG AAG TCT AGG GC) specific for *P. maniculatus* [49]. QuantStudio Design and Analysis Software v1.5.1 and GraphPad Prism v9.0.2 were used for analysis and graphs.

### Dryad Doi

10.5061/dryad.2547d7x2c [50].

### Live SNV isolation from tissue samples

**Cell culture.** Primary deer mouse lung endothelial cells (DMCD31 or PMVEC) were a gift from Dr. Tony Schountz and Dr. Kartik Chandran [25]. Freshly prepared VascuLife EnGS-MV Medium Complete Kit (Lifeline Cell Technology–#LL-0004) was used for media. Cells (P7) were thawed in water bath at 37˚C for 1 minute, then re-suspended in 3 mL media. Cells were plated in 25 mL flasks and incubated overnight. Fresh media was replaced the next day. Based on confluency, cells were passaged either 48–72 hours into 75 mL flask using PBS and TrypLE Express Enzyme 1x (Gibco Cat. #1260510). 250 or 500 μl of resuspended cells were seeded on either 48- or 24-well plate, respectively then after 24 hours were subjected to infection.

**Tissue and feces homogenization.** 2.0 mL bead beater tubes were prepared with a mixture of 1.0 and 2.3 mm zirconia beads and 800 μl of sterile PBS. Tissue homogenates (lung, heart, and liver) each weighing 0.5 g were then loaded and beaten two times for 30 seconds each, waiting a minute in-between. Tubes were spun down at 500 rpm for 5 minutes. 200 μl of supernatant was collected from each tissue and combined, then stored until infection. Feces required a faster spin, of 10,000 rpm for 1 minute. Once the feces supernatant was obtained, a 1 mL sterile BD Luer-Lok syringe (Cat #309628) loaded up to 400 μl was used. Feces were then filtered using a single-use, 0.22 μm filter (GVS Lot #7078568). Additional PBS was then used to wash filter and beads.

**Infection by tissue homogenates and collection.** All infections were done in BSL-3 conditions. After PMVEC were seeded, VascuLife media was removed and either 150 μl or 300 μl of lysate was placed on top for infection for two hours. After removing lysate, cells were washed with sterile PBS. Once washed, 600 μl of fresh VascuLife media was added without the addition of other antibiotics, allowing for incubation overnight before checking for potential contamination. Collection of cells and supernatant were performed every 7 days and passaged on fresh uninfected cells. 140 μl of supernatant was collected in pre-loaded tubes of 560 μl AVL prepared with carrier RNA, while cells were collected in 350 μl RLT prepared with β-ME and stored in -80˚C.

### Supporting information

**S1 Fig. Pulmonary microvascular endothelial cells (PMVEC) deer mouse derived cells can be infected by SNV-positive tissue homogenates.** (A) Experimental design using tissue homogenates of heart, kidney and/or lung from screened SNV-positive rodents are used to deliver virus to PMVEC from deer mice lung tissue and detected through RT-qPCR. (B) Copies per mL of SNV for different tissue homogenates during infection at multiple time points.

No template control (NTC) and positive (Pos) control were used for threshold adjustment. Images created with Biorender.com.
(DOCX)

## Acknowledgments

We thank Mariel Campbell and the technicians from the UNM Division of Genomic Resources for supplying a portion of our samples. We thank the numerous undergraduate and graduate student field crews associated with the Museum of Southwestern Biology. We also thank Jason Malaney, Curator of Biosciences at the New Mexico Museum of Natural History & Science. We thank Dr. Tony Schountz (Colorado State University) for providing PMVECs for this experiment. We thank members of the Bradfute Lab field crew for their assistance (Frances Twohig, Tonilynn Baranowski, and Dr. Andrew Skidmore). Lastly, we like to thank BLM Roswell and Carlsbad offices and multiple private landowners for access to the study sites.

## Author Contributions

**Conceptualization:** Samuel M. Goodfellow, Steven B. Bradfute.

**Data curation:** Samuel M. Goodfellow, Robert A. Nofchissey, Jaecy K. Banther-McConnell, Thanchira Suriyamongkol, Joseph A. Cook, Jonathan L. Dunnum, Ivana Mali, Steven B. Bradfute.

**Formal analysis:** Samuel M. Goodfellow, Robert A. Nofchissey, Jaecy K. Banther-McConnell, Joseph A. Cook, Jonathan L. Dunnum, Ivana Mali, Steven B. Bradfute.

**Funding acquisition:** Ivana Mali, Steven B. Bradfute.

**Investigation:** Samuel M. Goodfellow, Robert A. Nofchissey, Chunyan Ye, Jaecy K. Banther-McConnell, Thanchira Suriyamongkol, Joseph A. Cook, Jonathan L. Dunnum, Ivana Mali, Steven B. Bradfute.

**Methodology:** Samuel M. Goodfellow, Robert A. Nofchissey, Chunyan Ye, Jaecy K. Banther-McConnell, Joseph A. Cook, Jonathan L. Dunnum, Ivana Mali, Steven B. Bradfute.

**Project administration:** Steven B. Bradfute.

**Resources:** Joseph A. Cook, Jonathan L. Dunnum, Ivana Mali, Steven B. Bradfute.

**Supervision:** Ivana Mali, Steven B. Bradfute.

**Validation:** Robert A. Nofchissey, Jonathan L. Dunnum.

**Writing – original draft:** Samuel M. Goodfellow, Robert A. Nofchissey, Jaecy K. Banther-McConnell, Thanchira Suriyamongkol, Joseph A. Cook, Jonathan L. Dunnum, Ivana Mali, Steven B. Bradfute.

**Writing – review & editing:** Samuel M. Goodfellow, Robert A. Nofchissey, Chunyan Ye, Jaecy K. Banther-McConnell, Thanchira Suriyamongkol, Joseph A. Cook, Jonathan L. Dunnum, Ivana Mali, Steven B. Bradfute.

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
