## [Decision Letter · Decision Letter 0]

17 Sep 2024

Dear Dr. Bradfute,

Thank you very much for submitting your manuscript "A human pathogenic hantavirus circulates and is shed in taxonomically diverse rodent reservoirs" for consideration at PLOS Pathogens. As with all papers reviewed by the journal, your manuscript was reviewed by members of the editorial board and by several independent reviewers. In light of the reviews (below this email), we would like to invite the resubmission of a significantly-revised version that takes into account the reviewers' comments.

We cannot make any decision about publication until we have seen the revised manuscript and your response to the reviewers' comments. Your revised manuscript is also likely to be sent to reviewers for further evaluation.

Sincerely,

Amy L. Hartman, PhD

Academic Editor

PLOS Pathogens

Thomas Hoenen

Section Editor

PLOS Pathogens

Michael Malim

Editor-in-Chief

PLOS Pathogens

orcid.org/0000-0002-7699-2064

Reviewer's Responses to Questions

**Part I - Summary**

Reviewer #1: The report by Goodfellow et al. provides important data to support an expanded view of the species involved in maintenance and transmission of hantaviruses. While evidence of hantavirus infection (predominantly by serology) has indicated that other host species are susceptible to infection, it was believed that maintenance and transmission of each hantavirus species was linked to one main host. This report has many strengths including, but not limited to the large sample set representing various years of sampling; including more recent samples and archive samples; a wide diversity of putative host species tested; evaluation of both tissues and other biological samples (feces and salivary glands); PCR analyses and infectious virus isolation; and analysis of data with consideration for epidemiology of human cases. The advances provided in this report are key to expanding knowledge of hantavirus ecology. The only weaknesses in this report are areas that could be revisited for clarity and readability which are described below in “Part III – Minor Issues” section of review. Importantly, this report would be strengthened by expanding table and figure legends with more details of the samples tested, sample sizes and making sure it is very clear which data represent RNA quantification alone vs. infectious virus.

Reviewer #2: This is a very large survey of Sin Nombre virus (SNV) in wild-caught small mammals in New Mexico. Sin Nombre is a bunyavirus in the genus orthohantavirus, that can cause fatal disease. The primary reservoir for SNV is the western deer mouse that is widely distributed throughout the United States. In New Mexico, the vast majority of the human cases occur in the northwestern part of the state. However, seroprevalence studies suggest SNV is present elsewhere in the state. Previous studies also suggested that other small mammal populations are positive for SNV. The authors found that 25% of the nearly 1600 lung samples were positive for SNV RNA and that many different mammal species were positive. Next they attempted to isolate the virus from 90 different samples from different species. After several passages, SNV RNA was detected in many different tissues and species.

This manuscript is an follow up on a paper published earlier this year in PLoS One by the same group describing the Distribution and prevalence of Sin Nombre hantavirus in rodent species in eastern New Mexico. In this study they tested for SNV RNA in over 700 specimens and concluded that SNV RNA could be detected in many different small mammals across the state, reducing some of the novelty of the work presented here. Aside from that, no virus sequence data is presented. Comparisons of sequences between different mammal species and locations in New Mexico may offer insights into why most of the human cases of SNV occur in the northwestern parts despite the presence of the virus in other parts.

**Part II – Major Issues: Key Experiments Required for Acceptance**

Reviewer #1: No major issues.

Reviewer #2: Gene sequence data needs to be included to (a) confirm that these are unique viruses and isolates that were expanded, (b) to add significance to the paper, and (c) identify species or location differences in the SNV genome sequence. Even just sequencing the envelope protein on all the cultured viruses and some of the original tissues with high virus loads would greatly improve this paper.

Can the authors explain how the virus in Figure 4 can go up and down for some of the isolates? And how does this compare to a control virus?

**Part III – Minor Issues: Editorial and Data Presentation Modifications**

Reviewer #1: Line 27: Please clarify what “25% frequency rate” refers to. Overall (all years/species sampled)? Average of mean at individual sites? The study is quite extensive and it’s not clear in abstract what the value encompasses.

Line 49: Can you please specify what the case count is to date in NM (vs. just saying “highest”).

Line 77: While sampling is detailed in M&M, it would be helpful to list briefly here also. For samples from 2019-2023 especially, as lung alone is specified for the archive samples. It would be helpful to specify that terminal sampling was performed.

Line 77-88: Please specify, if possible, if there were months/seasons where sampling was performed, or year-round?

Table 1: Please add a column with common names in addition to species. Common names are mentioned later in the paper, and it would be helpful to have them in this table. Also please clarify if the “# screened (% total)” include the archive or is independent. If independent and represents the newer samples, is it positive on one or more tissues? Lung alone? Would be helpful to clarify which tissues were considered for other column given lung is specified for last column.

Line 161: Does “SNV lung titer” refer to RNA or infectious virus? Not clear as written. Because assays are done to look at vRNA and others with intent to assess infectious virus, please review text and specify vRNA load/titer when infectious virus not assessed. It is confusing in some sections which data reflects infectious virus vs. RNA alone and this is a key aspect (and important advancement) provided in this work.

Figure 3: The authors do a great job of linking data to geography, but it would be helpful to have more transparency about sample sizes, as they vary (for example in 3D). Please add details of sample sizes to text and figure legends wherever possible in this figure, and throughout, to aid readers in interpretation of data and to increase utility for comparison with future studies.

Table 2: Please clarify the sampling dates in table legend, in addition to that noted in text. Also clarify if only vRNA assessed and if only in Lung (as opposed to table 1 which may include other tissues?). Location # and sites are listed. Does the report include a key or more info regarding where in the state these locations are? Include site number info in Figure 3 or in supplemental table?

Line 232-238: Developed isolation protocol will be of great use for the field. However, there is no section in M&M that clearly outline isolation protocol (cell culture mentions lung endothelial cells – should use PMVEC acronym here to be clear?) – also not clear what antibiotics/antifungals were added when attempting isolation from different sample types (specify if none used because this is important point).

Line 239: How were the 90 samples used to attempt virus isolation selected? Based on Ct values? Cut-off? representative of different species? Please clarify.

Reviewer #2: Review Fig S1. The arrow suggests cells were put into animals?

This reviewer also suggests to put the high SNV RNA positivity rate into context. For example, is SNV causing a crhonic or an acute infection in mice and other rodents? If acute, how do you explain this high rate of SNV positivity? Are the infected animals more likely to get caught? Or does it cause a chronic life long infection?

PLOS authors have the option to publish the peer review history of their article (what does this mean?). If published, this will include your full peer review and any attached files.

Reviewer #1: No

Reviewer #2: No
---

## [Editor Report · Decision Letter 1]

20 Dec 2024

Dear Dr. Bradfute,

We are pleased to inform you that your manuscript 'A human pathogenic hantavirus circulates and is shed in taxonomically diverse rodent reservoirs' has been provisionally accepted for publication in PLOS Pathogens.

Best regards,

Amy L. Hartman, PhD

Academic Editor

PLOS Pathogens

Thomas Hoenen

Section Editor

PLOS Pathogens

Sumita Bhaduri-McIntosh

Editor-in-Chief

PLOS Pathogens

orcid.org/0000-0003-2946-9497

Michael Malim

Editor-in-Chief

PLOS Pathogens

orcid.org/0000-0002-7699-2064
---

## [Editor Report · Acceptance letter]

5 Jan 2025

Dear Dr. Bradfute,

We are delighted to inform you that your manuscript, "A human pathogenic hantavirus circulates and is shed in taxonomically diverse rodent reservoirs," has been formally accepted for publication in PLOS Pathogens.

Best regards,

Sumita Bhaduri-McIntosh

Editor-in-Chief

PLOS Pathogens

orcid.org/0000-0003-2946-9497

Michael Malim

Editor-in-Chief

PLOS Pathogens

orcid.org/0000-0002-7699-2064